# On the Capacity and Superposition of Minima in Neural Network Loss Function Landscapes

## Abstract

Minima of the loss function landscape of a neural network are locally optimal sets of weights that extract and process information from the input data to make outcome predictions. In underparameterised networks, the capacity of the weights may be insufficient to fit all the relevant information. We demonstrate that different local minima specialise in certain aspects of the learning problem, and process the input information differently. This effect can be exploited using a meta-network in which the predictive power from multiple minima of the LFL is combined to produce a better classifier. With this approach, we can increase the area under the receiver operating characteristic curve (AUC) by around $20\%$ for a complex learning problem. We propose a theoretical basis for combining minima and show how a meta-network can be trained to select the representative that is used for classification of a specific data item. Finally, we present an analysis of symmetry-equivalent solutions to machine learning problems, which provides a systematic means to improve the efficiency of this approach.

## 1 Introduction

Deep learning with neural networks (NNs) is a high-dimensional, non-convex optimisation problem for a loss function landscape (LFL). The coordinates of a *minimum* in the LFL are a *set of weights* for the machine learning model and a locally optimal *solution* to the learning problem, and these terms will therefore be used interchangeably throughout. It follows that the coordinates of the global minimum of the LFL are the weights that produce the lowest possible value of the loss function for the training data. The aim of machine learning is usually for the model to find a set of weights that fit the training data, but also generalise well to unseen testing data. Our approach extends this view. Instead of looking at just one minimum of the LFL, we are interested in the expressive power of multiple minima. To analyse how different minima extract and process information from the input data, we survey numerous low-lying minima of the LFL. Here, we employ tools from the energy landscape approach (Wales, 2003) to gain new insight into machine learning LFLs (Ballard et al., 2017). We note that the concept of a minimum is somewhat abstract in machine learning landscapes compared to molecular systems. While in a molecular energy landscape only minima provide valid configurations for a stable molecule, this restriction does not apply to LFLs for machine learning. In fact, some low-lying non-minima will have a smaller loss value and higher classification accuracy than a high-lying minimum. Here, we are interested in developing a better understanding of the capacity of diverse minima of the LFL, and showing that by combining the expressive power of different minima, we can build a better classifier. The compact form of this predictor provides a balance between accuracy and efficiency as required in applications where evaluation is a computational bottleneck.

### 1.1 Background

Machine learning models are structurally limited in the amount of data they can fit: their capacity is finite. The most commonly known measure of capacity is perhaps the Vapnik–Chervonenkis (VC) dimension (Vapnik & Chervonenkis, 1971; Vapnik et al., 1994). The higher the VC dimension, the more complex are the data can be fitted. More rigorously, VC dimension is defined as the largest cardinality of a set of data points that the NN can shatter (for our purpose, shatter means classify correctly). Thus, the weights of an underparameterised model (i.e. fewer parameters than training data points) may be incapable of fitting the entire test data set, but instead fit just parts of it.

The approach we employ to study the expressive power of combinations of individual minima is a variation of ensemble learning, where the results of multiple different predictors are combined to improve the overall accuracy of an approximation problem (Dong et al., 2020). The idea of combining multiple sources of information, specifically the output predictions of multiple classifiers, has been considered for over two decades (Breiman, 1996; Hashem, 1997; Jin & Lu, 2009). Two of the most important questions in ensemble learning are: which classifiers to consider, and how to combine the individual predictions (Wang, 2008). For a detailed review see Kuncheva (2014).

## 1.2 MOTIVATION

In this contribution, we are interested in quantitatively and systematically characterising a cornerstone of ensemble learning, namely classifier diversity. Logically, ensemble learning works if different classifiers extract different information from the input data or process it differently (Melville & Mooney, 2005; Zaidi et al., 2020). In the present work, the classifiers in question correspond to local minima of a reference neural network. We aim to visualise diversity of minima from the corresponding LFL and show how to select a few of them to produce a compact yet more accurate classification. We will show that different minima of the LFL successfully classify distinct subsets of the entire input dataset. Hence different local minima specialise in distinct parts of the test dataset, which we believe has not been shown before.

In summary, our main contributions are:

- Showing that different local minima specialise in distinct subsets of the input
- MLSUP, a proof-of-concept method that exploits minima diversity to improve classification results for complex problems
- An interpretation of the limitation of single-minima models and visualisation of the differences between minima
- Novel insights into the symmetry properties of minima in neural network LFLs

## 2 SUPERPOSITION OF MACHINE LEARNING SOLUTIONS: MLSUP

We observe that different local minima of a reference neural network extract different information from the input data and that combining just a few examples can improve classification significantly. To study this effect, we employ a modified stacking approach where multiple minima from the same classifier are combined, rather than multiple classifiers. We do not obtain these minima by different random initialisation but rather from sampling solutions from the LFL. This approach provides insight into the functional landscape and a deeper understanding of LFL minima. To answer the second important question in ensemble learning design, we employ a second neural network to select one of the local minima for a given input data item. This idea is related to previous theory, Jordan & Jacobs (1994) where a gating network chooses which classifier to apply to some problem (Shazeer et al., 2017; McGill & Perona, 2017). We call our method MLSUP, to denote a superposition of machine learning solutions (local minima of the LFL). We describe MLSUP in a four step process (Figure 1). The first step involves characterising local minima $\mathcal{M}$ by exploring the loss function landscape during training. Next, we choose a subset of minima $\mathcal{M}' \subseteq \mathcal{M}$ and evaluate each $m \in \mathcal{M}'$ for every training datapoint, which reveals how well each of them can classify specific data items (step 2 in Figure 1). A detailed discussion of how a few minima are selected for combination is included below. The superposition of chosen minima is done by training a second, meta-network (classifier 2, i.e. step 3 in Figure 1) to learn which of the $m \in \mathcal{M}'$ minima is best suited to classify a specific input datapoint. Thus, the second network learns to apply different minima to classify different types of input data, as shown in Step 4 of Figure 1. A pseudocode version of MLSUP is provided in the Appendix.

## 3 MODEL

We consider a classification problem for $C$ classes with a single hidden layer, as we are specifically interested in underparameterised networks. For some data $\mathcal{D} = (\mathbf{X}, \mathbf{c})$, the inputs are denoted $\mathbf{X} = \{\mathbf{x}^1, \ldots, \mathbf{x}^N\}$, where $N$ is the number of data points in the training or testing set, which are denoted as $\mathbf{X}_{train}$ and $\mathbf{X}_{test}$ respectively. The correct label for some data point $d$ is defined as $c^d$. We

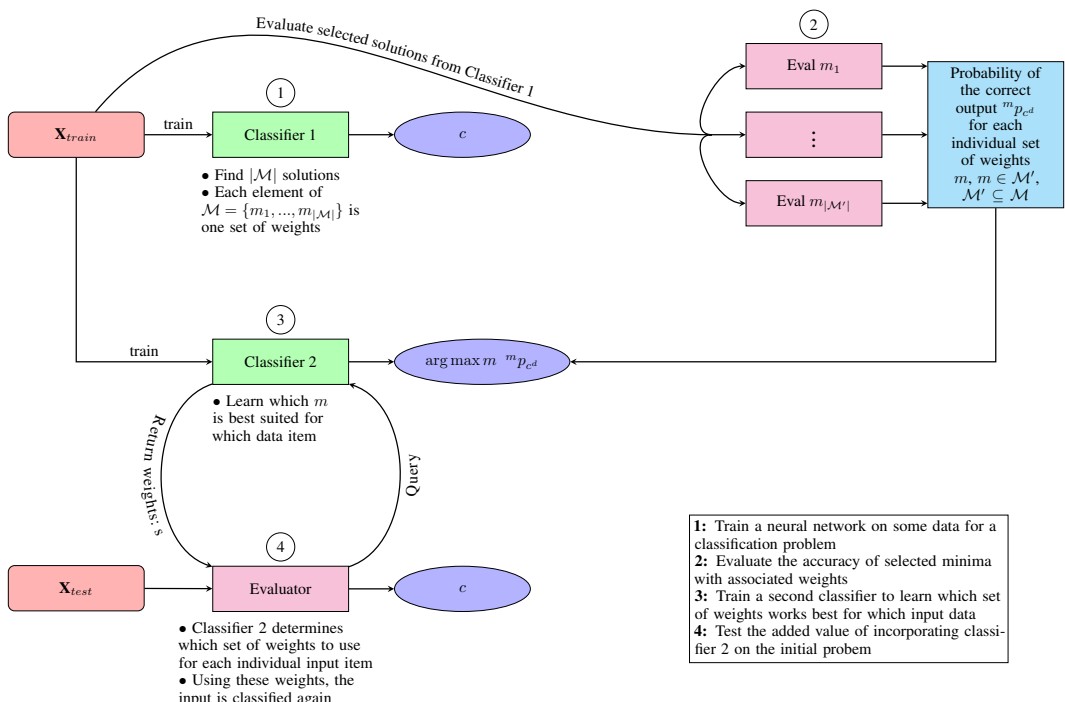

Figure 1: Diagrammatic representation of the MLSUP learning procedure. Ellipsoid boxes contain the true outputs, which are compared to the predicted output for training/evaluation purposes.

use $\tanh$ as the nonlinear activation function to the hidden layer, since it has continuous derivatives, which we require for optimisation. Outputs at node $y_i$ are converted to softmax probabilities $p_i(\mathbf{W}; \mathbf{X}) = \exp(y_i) / \exp(\sum_j e^{y_j})$, where $\mathbf{W}$ denotes the vector containing all weights. During training, we minimise a loss function $L(\mathbf{W}; \mathbf{X})$ with respect to these weights. We use a cross-entropy loss function

$$L(\mathbf{W}; \mathbf{X}) = -\frac{1}{N} \sum_{d=1}^{N} \ln p_{c^d}(\mathbf{W}; \mathbf{X}) + \lambda \mathbf{W}^2 \tag{1}$$

where $c^d$ is the correct class for some data item $\mathbf{x}^d$. A L2 regularisation term $\lambda \mathbf{W}^2$ is added to eliminate zero Hessian eigenvalues, which by Noether's theorem arise as a consequence of continuous symmetries in the loss function when additively shifting all the output bias weights (Ballard et al., 2017). We find $\lambda = 10^{-5}$ to be appropriate for the tests considered; our conclusions are largely insensitive to $\lambda$. This setup is used for the neural networks in steps 1, 2 and 4 of Figure 1.

### 3.1 DEFINING THE META-NETWORK LOSS FUNCTION

For step 3 of Figure 1, the loss function is different from equation 1. This section describes Classifier 2 from Figure 1, which is distinct from the one used for steps 1, 2 and 4. For Classifier 2, we are not interested in learning $c^d$, i.e. the correct output class, but rather the best local minimum to classify some input data item $d$, which is defined by the highest corresponding probability:

$$b^d = \arg\max m \ \ ^m p_{c^d} \ \ \forall \ m \in \mathcal{M}' \tag{2}$$

for data item $d$, where $m$ is one set of weights. This formulation changes the loss function to

$$L(\widetilde{\mathbf{W}}; \mathbf{X}) = -\frac{1}{N} \sum_{d=1}^{N} \ln p_{b^d}(\widetilde{\mathbf{W}}; \mathbf{X}) + \lambda \widetilde{\mathbf{W}}^2. \tag{3}$$

with $\widetilde{\mathbf{W}}$ representing the weights for network 2. We evaluate the classification predictions of our model using the area under the receiver operating characteristic curve (ROC-AUC) (Fawcett, 2006). The change in loss function also impacts the way we calculate the AUC for Step 4 in Figure 1. In the usual case, the AUC is given as

$$\text{AUC} = \int_0^1 T(P)dF(P) \tag{4}$$

with the true positive rate for outcome number one, $T(P)$, and the false positive rate, $F(P)$,

$$T(P) = \frac{\sum_d^{N_{\text{data}}} \delta(c^d - 1)\Theta(p_1 - P)}{\sum_d^{N_{\text{data}}} \delta(c^d - 1)} \qquad F(P) = \frac{\sum_d^{N_{\text{data}}}[1 - \delta(c^d - 1)]\Theta(p_1 - P)}{\sum_d^{N_{\text{data}}} 1 - \delta(c^d - 1)} \tag{5}$$

where $\delta(c^d - 1)$ is the Dirac delta function, and $\Theta(p_1 - P)$ the Heaviside step function, defined as

$$\delta(c^d - 1) = \begin{cases} 1 & \text{if } c^d = 1, \\ 0 & \text{if } c^d \neq 1 \end{cases} \qquad \Theta(p_1 - P) = \begin{cases} 1 & \text{if } p_1 \geq P, \\ 0 & \text{if } p_1 < P \end{cases} \tag{6}$$

However, we now have $|M'|$ possibilities. Thus, we must evaluate the AUC using the minimum $s^d$ that is chosen by Classifier 2 from Figure 1

$$T(P) = \frac{\sum_d^{N_{\text{data}}} \delta(c^d - 1)\Theta(p_1^{s^d} - P)}{\sum_d^{N_{\text{data}}} \delta(c^d - 1)} \qquad F(P) = \frac{\sum_d^{N_{\text{data}}}[1 - \delta(c^d - 1)]\Theta(p_1^{s^d} - P)}{\sum_d^{N_{\text{data}}} 1 - \delta(c^d - 1)} \tag{7}$$

## 3.2 OPTIMISATION ROUTINE

To survey the loss function landscape we employ methods from the energy landscape approach, which has been widely used to study molecular and condensed matter systems in the physical sciences (Wales, 2003). Specifically, global optimisation is performed using the basin-hopping method (Li & Scheraga, 1987; Wales & Doye, 1997) with a customised quasi-Newton L-BFGS (Nocedal, 1980) optimiser. More information is included in the Appendix. Candidates for transition states are obtained using a doubly-nudged (Trygubenko & Wales, 2004a;b) elastic band (Henkelman & Jónsson, 2000; Henkelman et al., 2000) approach and accurately refined by hybrid-eigenvector following (Munro & Wales, 1999; Zeng et al., 2014). The routines are implemented in the GMIN GMI, OPTIM OPT and PATHSAMPLE PAT programs, which are available under the GNU General Public License.

## 4 MINIMA SELECTION

The problem analogous to classifier selection in ensemble learning for MLSUP is minima selection. Here our motivation is to identify a small number of minima that significantly enhance the accuracy of the predictions, to produce a compact representation suitable for use in large-scale simulations that are typical in physical science applications. Hence we need to identify complementary minima that can be combined to improve the classification. We compare two methods for minima selection: one based on the theory of a thermodynamic analogue for the LFL, the other on more abstract machine learning concepts. We will also explain why simply picking minima that are 'far away' in Euclidean distance can be ineffective.

## 4.1 EUCLIDEAN DISTANCE

The Euclidean distance between two weight vectors is simply $|\mathbf{w}_1 - \mathbf{w}_2| = \sqrt{(\mathbf{w}_1 - \mathbf{w}_2)^2}$. However, minima 'far away' from each other in Euclidean space may not extract different information from the training data. In fact, there are symmetry-related minima, potentially distant in Euclidean space, that for the same input data will return exactly the same loss value. We refer to these solutions as permutational invariants or isomers, as for molecular systems. This sort of problem was considered in Dinh et al. (2017), where redundant minima are shown to exist for non-negative homogeneous nonlinear functions, such as the rectifier function $\phi_{rect}(x) = \max(x, 0)$. However, these results do not apply here because $\tanh$ is not homogeneous. Instead, let $\mathcal{G}$ denote the permutation group for some set of weights $w$, i.e. all permutations allowed such that for the same input, they return the same

output. By shuffling hidden nodes (Brea et al., 2019), and the fact that tanh is an odd function, we can show that for each minimum, there exist

$$|\mathcal{G}| = \prod_{l=1}^{H}(n_l! \times 2^{n_l}) \quad \text{or} \quad |\mathcal{G}| = \prod_{l=1}^{H-1}(n_l! \times 2^{n_l}) \times n_H! \tag{8}$$

degenerate solutions for each minimum, depending on whether the number of hidden layers $H$ is odd or even, respectively, where $n_l$ is the number of nodes in hidden layer $l$. Because tanh is an odd function, we can change the sign of the weights before and after each hidden node and each combination of any number of nodes for an odd number of hidden layers. This result increases the per-layer number of degenerate minima from $n!$ to $n! \times 2^n$, i.e. for each node swap, any nodes in each element of the power set of the nodes $\mathbb{P}(n)$ could be multiplied by a factor of $-1$ with no effect on the output. Thus, randomly picking some 'far away' minima might just return permutationally invariant solutions, which does not lead to any performance improvement in MLSUP. For further interest, we have included in the Appendix an example of four such permutational invariants and a pairwise distance matrix of permutational isomers of one minimum. In conclusion, a large Euclidean distance for minima selection is neither a necessary, nor a sufficient selection condition.

## 4.2 MISCLASSIFICATION SPACE

Instead of computing the distance between two sets of weights in Euclidean space, we can compute a metric in misclassification space. The rationale behind this approach is that minima with a high pairwise misclassification rate will be complementary, and therefore good candidates for MLSUP. We denote a vector containing the predicted output class for each data item $i \in X$ for some minimum $m$ as $\mathbf{c}_i^{(m)}$, and define the set $\Xi = \{n \mid \mathbf{c}_n^{(1)} \neq \mathbf{c}_n^{(2)}\}$, which contains the indices of all elements that are classified differently by the two minima. To obtain a misclassification ratio $\Lambda$ from this set, we simply represent the cardinality of $\Xi$ as a fraction of the number of data points $\Lambda = |\Xi|/N$, i.e. the number of misclassified items divided by the total number of items. If we compute $\Lambda$ for all $m \in \mathcal{M}$, where $\mathcal{M}$ again denotes the set of all known minima, we obtain a symmetric matrix of pairwise differences in classification space. We can then train MLSUP on selected minima that are distant in classification space. Below, we will abbreviate the Maximum Misclassification Distance to the Best Minimum as MMDBM. A more theoretical basis for weighting classifiers according to correlation of errors is given in Masegosa et al. (2020).

## 4.3 HEAT CAPACITY ($C_V$)

Building on results for molecular systems, we can compute a theoretical analogue of the heat capacity ($C_V$) (Wales, 2017). The heat capacity reports upon the changing occupation probabilities of local minima as the temperature changes, which depends on both the relative loss function value for the minimum, and a local density of states (the analogue of entropy) in weight space. By computing the partial derivatives of the occupation probabilities with respect to temperature, we can identify the minima with the largest rate of change of probability, which theory shows make the largest contribution to the heat capacity.(Wales, 2017) Peaks in the heat capacity correspond to changes in occupation between local minima with different loss function values and entropy. Thus, in physical systems, the minima with the largest positive and negative partial derivative around a peak generally have qualitatively different properties. Peaks in $C_V$ therefore tell us where to find local minima with complementary properties (Wales, 2017), which may provide good candidates for combinations in a new classifier. This reasoning as the original motivation for the MLSUP procedure, since the heat capacity analysis is computationally efficient and physically insightful.

## 4.4 DATA

To test MLSUP, we chose to study the spiral data problem (Lang & Witbrock, 1988). An example can be seen in Figure 4. This problem is considered to be relatively difficult because of its high degree of non-linear separability. We further increase the difficulty by adding a small uniform noise term to each datapoint. Our models are trained on 3,200 data points and tested on 800. To show that the MLSUP approach is generalisable, we report results for other datasets in the Appendix.

## 5 RESULTS

In this section, we report general results for MLSUP and specifically the impact on the AUC for different minima selection methods and different numbers of minima combined in MLSUP. We benchmark the MLSUP AUC in four ways. First, we compare it to the best individual AUC of the minima that we choose for MLSUP. Second, we compare it to the best AUC of the non-MLSUP system (classifier 1 in Figure 1) from which the MLSUP minima are selected. Third, in order to evaluate MLSUP, we compare its AUC with the best possible AUC achievable from combining the selected $n$ minima. To compute this theoretical maximum AUC (TMA), for each datapoint we select the maximum Softmax probability of the correct class amongst all minima considered. The TMA gives an upper bound for the accuracy of MLSUP. Thus, to evaluate the quality of Classifier 2 alone, an AUC achieved with MLSUP should be compared against the TMA. Lastly, we compare it with a simple majority vote (with ties considered as correctly classified).

### 5.1 MLSUP EFFECT

A substantial increase in the AUC can be achieved with MLSUP. For the original network (Classifier 1 in Figure 1), we found 7,842 minima and a maximum AUC value of 0.76. Using ML-SUP with two of these minima (6436 and 656), we obtain an AUC of 0.91. By combining four minima, the AUC increases to 0.94 (Table 1). The weights for these four minima are shown in Figure 2. There is no apparent relationship between these four sets, yet they are all minima of the original LFL and have similar AUCs. We also find that MLSUP outperforms a ma-

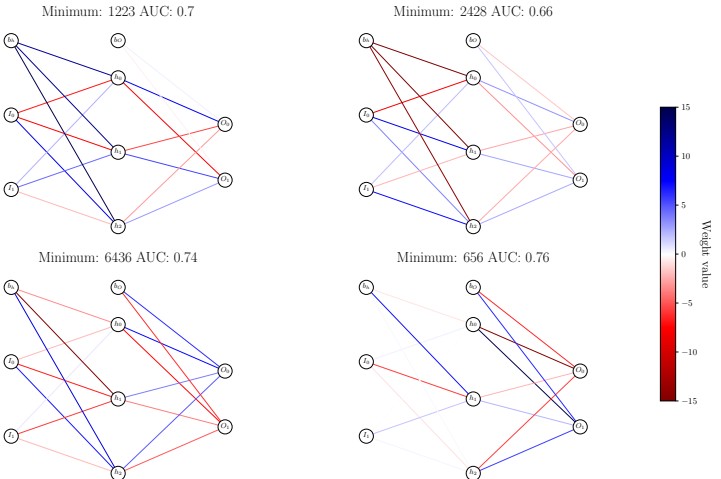

Figure 2: Four sets of weights, shown for a small network. All four minima have very different sets of weights, as evidenced by the colours. However, their AUC values for the test data are similar.

jority vote and include results for other benchmark datasets, higher-dimensional problems and MLSUP hyperparameters in the Appendix.

Table 1: AUC improvements for different MLSUP settings

| Selection method | # minima | Best individual | MLSUP | TMA | Majority vote |
|---|---|---|---|---|---|
| $C_V$ peak 1 | 2 | 0.70 | 0.88 | 0.98 | 0.76 |
| $C_V$ peak 2 | 2 | 0.70 | 0.86 | 0.97 | 0.73 |
| MMDBM | 2 | 0.76 | 0.91 | 0.97 | 0.78 |
| MMDBM & $C_V$ peak 1 | 4 | 0.76 | 0.93 | 0.99 | 0.86 |
| MMDBM & $C_V$ peak 2 | 4 | 0.76 | 0.94 | 0.99 | 0.85 |
| Max Euclidean distance | 2 | 0.65 | 0.67 | 0.68 | 0.66 |

### 5.2 MINIMA SELECTION

Table 1 clearly shows why a minima selection routine is necessary. Choosing the two minima with the largest Euclidean distance, we obtain an AUC of 0.67, substantially lower than for either of the systematic minima selection methods. Furthermore, the Pearson correlation coefficient between

distance in Euclidean and misclassification space is only 0.06, which demonstrates that Euclidean distance alone is not useful for minima selection. Additional details can be found in the Appendix.

### 5.2.1 MISCLASSIFICATION SPACE

The distance between minima in misclassification space is the best minima selection method for an MLSUP meta-network based on two minima. By combining the best individual AUC minimum (656) with the minimum furthest away from it in misclassification space (6436), we obtain an AUC value of 0.91 (MMDBM). We note that the individual AUC values of both these minima were reasonably high already, at 0.74 and 0.76. The best possible AUC that MLSUP could have achieved, if for each data item the better minimum had been chosen for this system (TMA) is 0.97 (Table 1).

### 5.2.2 HEAT CAPACITY

The heat capacity ($C_V$) curve for the LFL has two peaks, a small one at low $T$ and a larger peak at $T \approx 3 \times 10^{-3}$ (Fig. 3). Such a $C_V$ curve in a molecular system would indicate a solid-solid phase transition at low temperature (first peak), and a solid-liquid transition (melting) at higher temperature (second peak) (Wales, 2003). In the neural network case, these peaks can be understood in terms of transitions between two energetically/entropically different sets of minima. By combining the minima that make the greatest contributions to $C_V$ at the larger peak (Fig.3) in MLSUP, we improve the AUC to 0.86 (Table 1). For the minima that produce the smaller peak, the improvement is similar, with an AUC value of 0.88 (Table 1). The TMA of MLSUP is 0.98 for the first and 0.97 for the second peak.

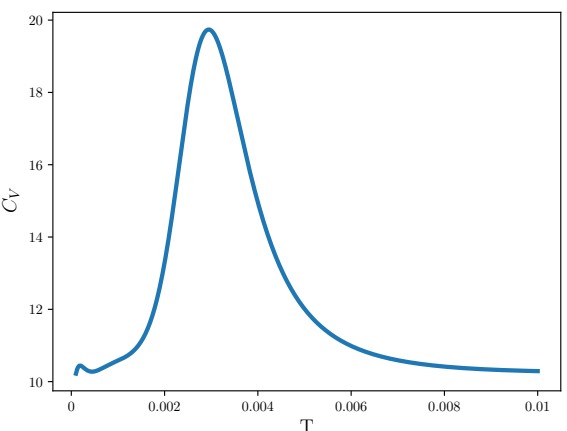

Figure 3: Heat capacity ($C_V$) curve for ML landscape. As the temperature is increased, the occupation probabilities change and more minima become accessible. $C_V$ is calculated using an analogue of normal mode analysis for each minimum (Bogdan et al., 2006).

## 6 DISCUSSION

Combining different minima in one classifier can substantially improve the ROC-AUC for a complex problem like spiral data. This result strongly suggests that different minima extract different information from the input and/or process it in a different way. Our results shed new light on the LFLs of complex problems. Even the global minimum is limited in the amount of information it can extract and process. To maximise the efficiency of MLSUP, we therefore aim to identify complementary high-AUC minima for which the intersection of correctly classified datapoints is small. This selection translates to a large distance in misclassification space. Such minima are ideal candidates for MLSUP.

### 6.0.1 COMPUTATIONAL RESOURCES

Training MLSUP for four minima is fast and hence does not constitute a bottleneck. All tasks were run on six-core dual Xeon X5650 (2.6GHz) nodes with 24Gb RAM per node. Using just one core, MLSUP is trained within a few hours. The initial analytical landscape exploration requires more subtle considerations. Due to the nature of the LFL (theoretically infinitely large, yet implicitly bounded by L2-regularisation), more minima can generally be found. For reference, finding 7,842 minima took us around 92 hours. The computing time required for landscape exploration depends on system size, number of data points, and various parameters in the optimisation routine, but also the structure of the LFL, which is specific to a given problem. Landscape exploration can be parallelised, so there is scope to speed up the calculations significantly in the future.

## 6.1 Applications for MLSUP

We emphasise that our main objective in the present contribution is to improve our understanding of the LFL and investigate whether the MLSUP approach might be useful. In particular, we have investigated the underlying assumptions concerning the information extracted by different minima for underparameterised networks. For larger, overparameterised networks, a single minimum may suffices to achieve near-perfect AUC. However, evaluating a complex classifier could become a computational bottleneck in large-scale simulations of physical systems. Hence, a compact MLSUP representation could be useful for such applications. For example, we have an application where predicted values of specific molecular properties are required at every step in a molecular simulation (Vendruscolo & Dobson, 2005; Dedmon et al., 2005; Lindorff-Larsen et al., 2005; Clore & Schwieters, 2006; Cavalli et al., 2007). The properties depend on the atomic coordinates, which vary throughout the simulation, and we need analytical derivatives of the properties (outputs) with respect to the coordinates (inputs). A compact but computationally efficient prediction engine, such as MLSUP, is needed to prevent these computations from becoming a bottleneck in the simulations.

## 6.2 Minima selection

Selecting the correct minima for MLSUP is a key problem. Even when combining just two out of $|\mathcal{M}|$ minima, there are $\binom{|\mathcal{M}|}{2}$ possible combinations. While a choice guided by Euclidean distance may be appealing, this approach is ineffective due to low misclassification distance or permutationally equivalent solutions. Hence, we have analysed two other methods, one based on a theoretical analogue of the heat capacity, and the other on misclassification space. Combining minima of high misclassification distance achieves the best results. However, evaluating the misclassification space is relatively expensive and scales $\mathcal{O}(n^2)$, where at each step, all data points must be evaluated for one of $n$ minima. In contrast, the analysis of $C_V$ scales linearly as $\mathcal{O}(n)$. Additionally, a high misclassification rate between two individually poor minima is unlikely to substantially improve results, as the confidence in each minimum would be low, and a superposition would not necessarily improve this situation. We have not observed any such issues with the $C_V$ approach. The superior results of $C_V$ as opposed to random initialisation are striking, and demonstrate how a methodology developed for molecular energy landscapes can provide new insight into machine learning LFLs.

## 6.3 Interpretation of differences between minima

Figure 4 provides an explanation for the substantial improvements that are achieved in MLSUP. While Figure 2 shows that the individual minima are very different from each other, it does not show how these differences are propagated to classification decisions. Figure 4 shows that individual minima are good at predicting specific ranges of the input data, and only in combination do they manage to classify points all around the spiral. Note that the Softmax probability cannot generally be read as true probability because it lacks uncertainty quantification (Sensoy et al., 2018). However, it can be used here as a visualisation proxy due to the well-behaved nature of the synthetic data. We see that the number of large points per minimum correlates with the AUC. Minima 6436 and 656 have AUCs of 0.74 and 0.76, respectively. In the right panel of Figure 4, large parts of the MLSUP predictions are red or blue, corresponding to these two minima. In contrast, minimum 2428, which only seems to be able to accurately predict datapoints from Class 1 in the upper right quadrant, has a lower AUC (0.66). Another feature clearly illustrated in Figure 4 is the input-sign dependence of the minima. In particular, minima 1223 and 2428 only have high confidence for datapoints that lie in one of the four segments. While 2428 is good at classifying Class 1 datapoints with positive sign for both inputs, minimum 1223 works best for classifying Class 0 datapoints with positive x and negative y. Thus, if MLSUP receives an input data point where both coordinates are positive (and perhaps at least one of them large), it is likely to classify it using the weights from minimum 2428.

Combining the above results with Figure 2, we see that minimum 6436 has strongly positive weights going into the upper output node $O_0$ and strongly negative weights into $O_1$, while the opposite holds for minimum 656. Indeed, Figure 4 confirms that 6436 is very good at classifying data belonging to output class 0, while 656 is very good at classifying data belonging to output class 1. We therefore conclude that different minima truly 'specialise' in parts of the input data, thus providing the foundations for the MLSUP approach. Visualising the results we provides an intuitive way to confirm that different minima process the same information in a different way. This observation raises various

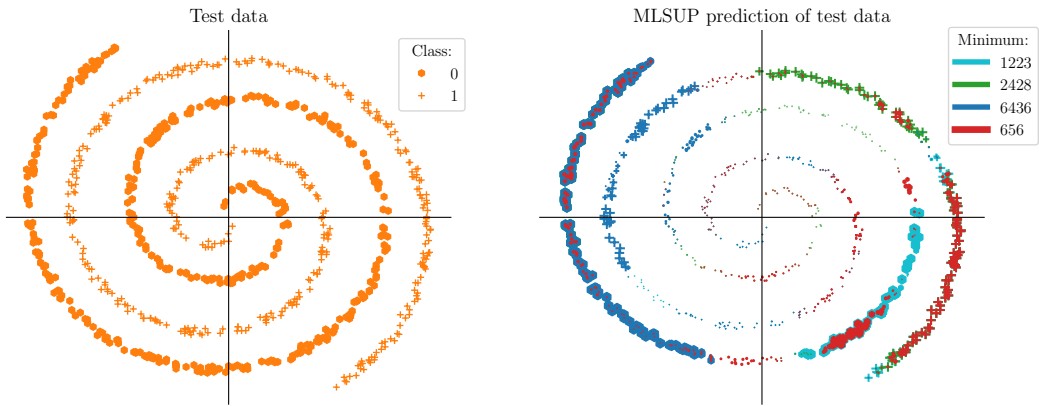

Figure 4: Contribution of each of four selected minima (right) to the MLSUP prediction of binary test data (left). Marker shape indicates the label, colour of the respective minimum, and size the Softmax probability that the respective minimum obtains for the correct class of this datapoint (scaled quadratically). Only points that are classified correctly by the respective minimum are included.

questions around initialisation methods and interpretability of ML models. Instead of considering model interpretability, it may be more appropriate to discuss the interpretability of different minima, as various solutions may extract, and hence focus on, completely different information from the input data. The overparameterisation of neural networks (Belkin et al., 2019) may explain why different minima achieve such good results individually. However, to explain neural network decision making, understanding that different minima process the input differently seems a crucial first step.

### 6.4 CONCLUSIONS

Different sets of weights associated with different minima of the loss function landscape extract different information from the input data, or process it differently. By training a meta-network to decide which minimum should be applied for any specific input data point, we can substantially improve classification accuracy. These results explain why different solutions to the same learning problem, obtained with different initialisation methods, have varying degrees of accuracy. Furthermore, we show that if the capacity of the network is limited, individual minima specialise in parts of the network. In future work, it will be interesting to consider these results in the light of mathematical measures of network capacity, such as VC-dimensions.

In addition to this insight, we also provide an analysis of redundant minima which, although they might appear to be different, return the same loss value for some input. Such symmetries make minima selection more challenging. Simple measures, such as the Euclidean distance in weight space, are insufficient due to low correlation with misclassification distance. To solve this problem, we can exploit thermodynamic analogues for the LFL by analogy to molecular systems. In particular, we find that defining a heat capacity analogue for the LFL provides an efficient and direct approach for locating complementary local minima. Computing the heat capacity, and identifying the minima that make the largest contributions to the peaks, is a promising method for constructing a compact but relatively accurate classifier in the MLSUP approach. This representation may prove useful for applications where evaluation of the classifier is a potential bottleneck, as for simulations in the physical sciences, where predictions of molecular properties are required at every step.Vendruscolo & Dobson (2005); Dedmon et al. (2005); Lindorff-Larsen et al. (2005); Clore & Schwieters (2006); Cavalli et al. (2007) The analogy to physical systems provides further motivation for pursing the energy landscape view of machine learning loss function landscapes.

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

## 7 APPENDIX

### 7.1 FURTHER EXPERIMENTS

Table 2 includes some additional results that reinforce our conclusions. Importantly, we also want to test MLSUP on other datasets to provide evidence that the performance increase is not specific to the spiral dataset. We first considered the well known Iris flower dataset? and, to make it more difficult, mislabel 45% of the training samples. The second additional test for MLSUP was for the checkerboard dataset, another synthetic, nonlinear problem. Similar to Iris, we achieve significant improvements using MLSUP. When the number of minima is increased substantially (e.g. to 20) the AUC only increases slightly. Training MLSUP for 20 minima takes around 10 times as long as for four minima using the same setup as above. These results again suggest that MLSUP can be effective for a small number of local minima, with the potential to provide useful accuracy at a low computational cost. This balance of accuracy and efficiency could be valuable in applications where reasonable predictions are required without a significant overhead.

Table 2: AUC improvements by MLSUP

| Description | # minima | AUC Best individual | MLSUP | TMA |
|---|---|---|---|---|
| Large hidden layer (n=100) | 2 | 0.99 | 0.99 | 1 |
| Small training set N=400 | 2 | 0.62 | 0.78 | 0.86 |
| 5 MLSUP minima | 5 | 0.76 | 0.94 | 0.99 |
| 20 MLSUP minima | 20 | 0.76 | 0.97 | 1 |
| 100 randomly chosen (mean) | 2 | 0.74 (0.67) | 0.7-0.89 (0.78) | 0.98 |
| 100 randomly chosen (mean) | 4 | 0.76 (0.67) | 0.78-0.93 (0.84) | 0.99 |
| Mislabelled Iris flower dataset | 2 | 0.85 | 0.96 | 0.99 |
| Checkerboard | 2 | 0.68 | 0.84 | 0.96 |

### 7.2 PERMUTATIONAL INVARIANTS

#### 7.2.1 EXAMPLES

Here we provide additional insight into the issue of permutationally equivalent sets of weights.

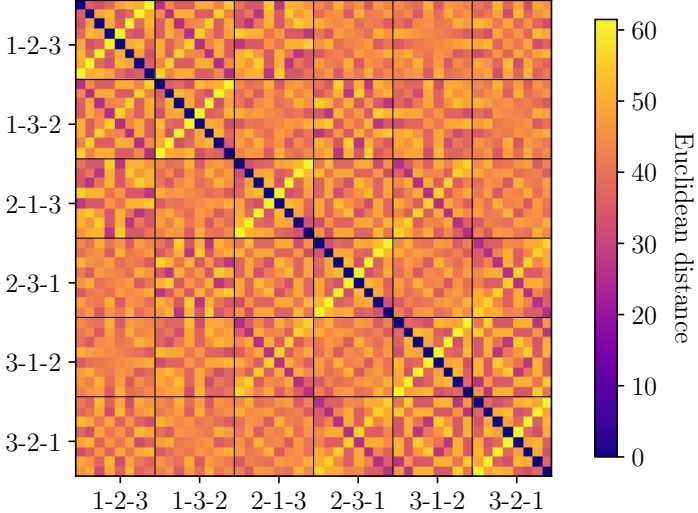

**Supplementary Figure** 5: Euclidean distance between 48 permutationally equivalent local minima of the global minimum for a particular LFL.

Figure 5 shows how distant permutationally equivalent local minima can be in Euclidean space. The system contains a single hidden layer with $n = 3$ nodes. Shuffling the nodes leads to $n! = 6$ permutations and within each permutation, the weights going in and out of any combination of the hidden nodes can be multiplied by $-1$. Thus, for each shuffle-permutation there are an additional $2^n = 8$ isomers, producing 48 in total. In misclassification space, the distance between all these 48 minima would be 0, as in Figure 6, which shows how far apart these equivalent solutions can appear to be.

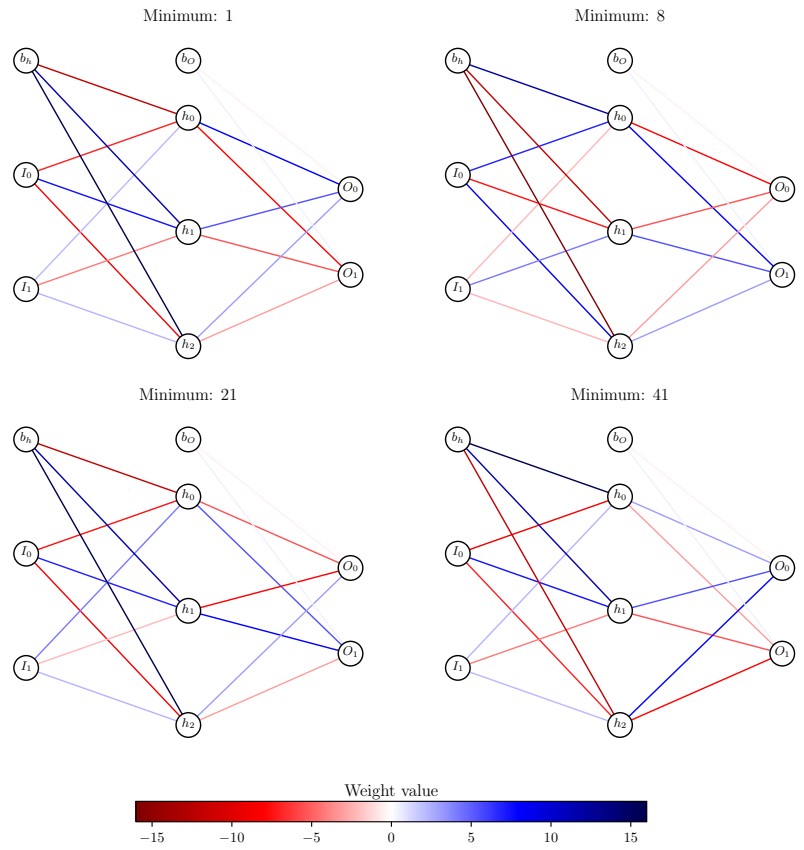

**Supplementary Figure** 6: An example of four very different looking minima that are actually permutationally equivalent, i.e. redundant solutions. For any input, these four networks will all return exactly the same loss value.

### 7.3    COMPARISON OF EUCLIDEAN AND MISCLASSIFICATION SPACE

This section presents further evidence that the Euclidean distance alone is insufficient for minima selection. Figure 7 shows that only a very weak correlation exits between Euclidean and misclassification distance. Large misclassification distance can be observed roughly as often for points of large Euclidean distance as it is observed for points close together in the parameter space defined by the weights. Given the good results of misclassification distance for minima selection, the low correlation coefficient indicates that Euclidean distance alone is not useful.

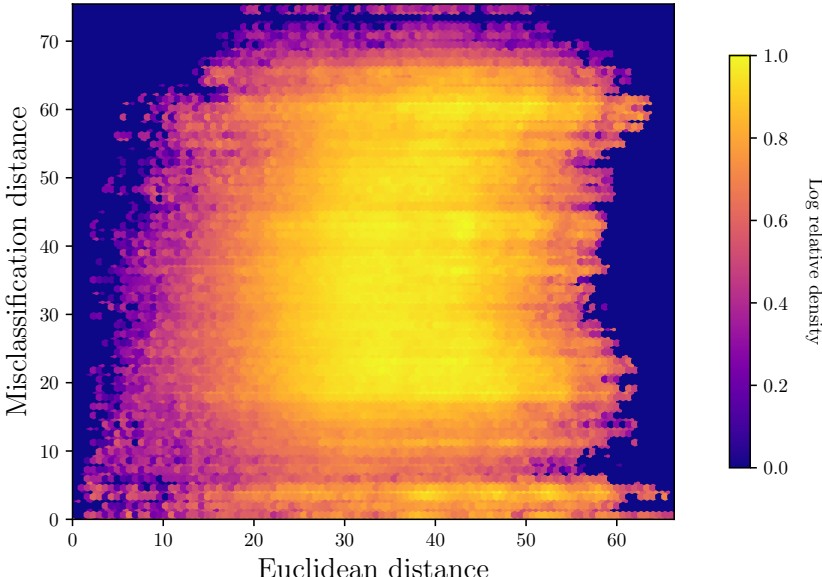

**Supplementary Figure** 7: Euclidean vs Misclassification distance for each of the $\binom{n}{2}$ combinations of $n$ minima of the spiral data LFL. The Pearson correlation coefficient for Euclidean distance vs misclassification is 0.06.

### 7.4   MLSUP ALGORITHM

The MLSUP algorithm can be represented in pseudocode as:

---
**Algorithm 1** Train MLSUP

---
**Require:** $\mathcal{D} = (\mathbf{X}, \mathbf{c})$
   Sample LFL, find $|\mathcal{M}|$ minima
   Pick some $|\mathcal{M}'|$ minima candidates
   Evaluate each minimum $m \in |\mathcal{M}'|$ on training data
   Train second nn to learn $\arg\max_m {}^m p_{c^d}$
   **for** $d \in \mathcal{D}$ **do**
      Evaluate first network, query second network for which minimum to use
   **end for**

---

### 7.5   CHOOSING $m'$ MINIMA BY PAIRWISE MISCLASSIFICATION DISTANCE

One may wish to compute MLSUP with $m'$ minima, chosen to have the maximum misclassification distance from another. This is a combinatorial optimisation problem, analogous to having $m$ cities with pairwise distances and trying to find the $m'$ cities that are furthest from each other. This problem can be solved by quadratic programming with convex relaxation. Let $\mathbf{D}$ be an upper triangular matrix with element $\mathbf{D}_{ij}$ denoting the misclassification distance from point $i$ to point $j$, where $i < j$. The square matrix $\mathbf{D}$ has 0 diagonal. Denote $x$ as a $m$-dimensional binary vector, which contains the value 1 at the position of a chosen minimum and 0 elsewhere. The objective is then to maximise $x^\top \mathbf{D} x$. In practice, the values of $x$ will be relaxed so they need not be discrete, and the optimisation procedure becomes

$$\max_y y^\top \mathbf{D} y \quad \text{subject to} \quad 0 \le y_i \le 1 \; \forall i, \; 1 \cdot y = m' \tag{9}$$

The final task is then to discretise $y$ back to $x$ such that $y^\top \mathbf{D} y$ is very close to $x^\top \mathbf{D} x$. We have achieved this mapping by simply setting the $m'$ largest elements of $y$ to 1 and all others to 0, but other methods of approximation may work equally well or even better.

### 7.6 BASIN-HOPPING GLOBAL OPTIMISATION

The first step in surveying the LFL for a neural network is to locate the global minimum. We perform global optimisation using the basin-hopping method, which uses a Metropolis criterion to avoid getting stuck in local minima (Li & Scheraga, 1987; Wales & Doye, 1997). For local minimisation we use a modified quasi-Newton L-BFGS (Limited-memory Broyden–Fletcher–Goldfarb–Shanno) algorithm (Nocedal, 1980). After a local basin of attraction (local minimum) is found via L-BFGS optimisation, a 'basin-hopping' jump is performed to some other point in the LFL. This jump is always accepted if the energy, here loss value, is lower than the current minimum. If the loss value of the new point is higher, it is accepted with probability

$$P \propto \exp\left(-\frac{\Delta \widetilde{E}}{k_{\mathrm{B}}T}\right) \tag{10}$$

where $\Delta \widetilde{E}$ is the difference in loss value between the current minimum and the new point, $k_{\mathrm{B}}$ the Boltzmann constant and $T$ a fictitious temperature. Intuitively, if the energy difference between the old and new points is large, the move is less likely to be accepted. A minimum is characterised as such if a RMS norm of gradient vector) convergence criterion at a threshold of $10^{-10}$ is reached. Each basin-hopping run naturally produces a sampling of low-lying minima as it progresses, in addition to the global minimum.

