# OpenReview forum: "On the Capacity and Superposition of Minima in Neural Network Loss Function Landscapes"
_ICLR.cc/2022/Conference — ICLR 2022 Submitted_

### Official Review · Reviewer_yRSp · 2021-10-22

**Correctness:** 2
**Technical Novelty And Significance:** 1
**Empirical Novelty And Significance:** 2
**Recommendation:** 3
**Confidence:** 4

**Main Review:**



## Pros
The motivation looks reasonable to me.

## Cons
The paper is poorly written and I find it hard to follow the authors' ideas. The mathematical notations look very strange to me. For instance, the authors use $d$ to denote the input; in Eq. (2), the left superscript denotes the variables to be optimized. Moreover, I do not understand the so-called heat capacity in Section 4.3 and Section 5.2.2. The authors should have provided detailed formulations and explanations in the appendix.

The proposed methods make sense only for under-parameterized networks. However, deep neural networks often work in the over-parameterized regime in modern applications. I would like to know if the proposed under-parameterized ensemble method outperforms well-tuned over-parameterized nets or not. In addition, the authors did not provide comparisons to other deep ensemble methods, e.g., Snapshot Ensembles: Train 1, get M for free (https://arxiv.org/abs/1704.00109). Therefore, it is not clear about the effectiveness of the proposed method.


## Other comments

1. For the abstrct,
    - LFL is introduced without providing the full name.
    - "by 20% for a complex learning problem". I do not think that the spiral dataset is complex given the application of modern deep learning. In particular, this paper aims to propose efficient applicable methods instead of theoretical analysis.
    - "We propose a theoretical basis ..." I did not see where the theoretical study is provided in this paper.

2. For Section 1.2:
   - MLSUP is first used without providing the full name. What is a "proof-of-concept method"?

3. For Section 2:
    - "We employ a modified stacking approach where multiple minima from the same classifier are combined, rather than multiple classifiers". I think that the combination of multiple minima has been extensively exploited in the deep learning community, e.g., the paper mentioned above.
    - What is a "functional landscape"?
4. For Section 3, Eq. (1), The mathematical formulation of the reguarlized risk seems wrong to me. $\mathbf{W}^2$ is not defined. $p_{c^d}(W;X)\to p_{c^d}(W;x_d)$.

5. Eq. (2) and Eq. (3) are strange.

6. I do not understand the purpose of Section 4.1. That the Euclidean distance is not a good metric to measure the difference in the high dimensional regime is well-known. I do not see any extra insight from this section.

7. In Algorithm 1, for the second steps, how do you pick $|\mathcal{M}'|$?



**Summary Of The Paper:**

## Summary
This paper proposes a new ensemble learning based on neural networks.  First, A basin-hopping method is used to sample diverse minima of the empirical risk landscape. Then, a meta-network is trained to predict which minimum performs the best for specific input. This meta-network maps the input sample to the index set of minima. This can be viewed as an adaptive ensemble method, in the sense that for different input samples, the evaluated models are different.

**Summary Of The Review:**

See above

---

### Official Review · Reviewer_saVg · 2021-11-02

**Correctness:** 2
**Technical Novelty And Significance:** 3
**Empirical Novelty And Significance:** 1
**Recommendation:** 3
**Confidence:** 5

**Main Review:**

Strengths:
- I like the general idea of having a companion network to choose the best network out of an ensemble.
- The two methods to select diverse minima are interesting and outperform the baselines.
- The results of the proposed method on the spiral dataset seem rather encouraging, but lack further support.
- The analysis of heat capacity looks like an interesting bridge with physics, but is poorly presented.

Weaknesses
- The spiral dataset considered is a very basic machine learning dataset, and is far from challenging enough to qualify as a proper deep learning task. The same goes for the Iris dataset used in the appendix. I understand that this dataset is convenient for visualisation of MLSUP (Fig. 4), and that not everyone can train on ImageNet due to computational constraints, but MNIST or CIFAR10 are a minimum to validate the method proposed.
- The paper lacks comparison with concurrent ensembling methods, such as snapshot ensembles, stochastic weight averaging, etc.
- The paper lacks clarity in parts. The physics jargon used in parts of the manuscript (heat capacity etc) is difficult to parse, I believe, for the machine learning audience. Sentences like “By computing the partial derivatives of the occupation probabilities with respect to temperature” need to be clarified or better, formalised mathematically. The Applications section 6.1 is also difficult to understand and does not add much value to the paper since the applications are not presented in the manuscript itself.
- The experiments lack description : what network is used (width, activations, depth etc) ? What is the optimiser ? etc.
- As acknowledged by the authors, MLSUP is effective for underparametrized networks, which specialise to parts of the data. However, Tab. 2 of the Appendix shows that MLSUP does not help for overparametrized networks. This is quite a strong limitation in my opinion, given that most networks used in practice are overparametrized, and should be stated more explicitly in the main text.

Comments :
- The paper is framed as a theoretical exploration of loss landscapes, but seems to me more applied, as most of the content is devoted to the MLSUP method. Perhaps the authors should change the title accordingly.
- “We use tanh as the nonlinear activation function to the hidden layer, since it has continuous derivatives, which we require for optimisation” : why ? This needs to be clarified.
- “Different local minima specialise in distinct parts of the test dataset, which we believe has not been shown before” : such broad claims should be avoided. Obviously, different minima do not perform exactly the same on different parts of test data. The extent to which they specialise entirely depends on the dataset used : the one considers here is very specific and favours strong specialization. The question of ensemble-of-specialist networks has also been studied extensively, see for example the Multiple Choice Learning line of literature [1,2,3].
[1] Guzman-Rivera et al 2012, Multiple Choice Learning: Learning to Produce Multiple Structured Outputs
[2] Lee et al 2016, Stochastic Multiple Choice Learning for Training Diverse Deep Ensembles
[3] Lee et al 2017, Confident Multiple Choice Learning

In the abstract :
- “that extract and process information from the input data to make outcome predictions” : this isn’t necessary
- “We demonstrate that different local minima specialise in certain aspects of the learning problem, and process the input information differently” : too vague
- “for a complex learning problem” : too vague
- “multiple minima of the LFL” : LFL needs to be defined


**Summary Of The Paper:**

This paper introduces a method for combining the expressive power of different minima of a given loss landscape, by using an auxiliary meta-network to select which minima is best suited to classify a given data point. They also introduce two methods to select diverse sets of weights : one based on misclassification distance, the other based on a heat capacity method from statistical physics.

**Summary Of The Review:**

Overall, I think this paper presents an interesting set of ideas, but strongly lacks empirical support. The experimental section is poorly explained and is too simplistic to support the method presented (the only dataset considered in the main text is a spiral dataset). Also, the paper is difficult to parse in parts, not because of mathematical formalism but because of the use of physics jargon with a lack of clarifications.
Therefore, in the current state, I would not recommend publication, but strongly encourage the authors to consolidate this interesting  research direction with more extensive experiments.

---

### Official Review · Reviewer_XRAh · 2021-11-02

**Correctness:** 3
**Technical Novelty And Significance:** 2
**Empirical Novelty And Significance:** 2
**Recommendation:** 3
**Confidence:** 4

**Main Review:**

This paper studies an interesting topic of classifier diversity in ensemble learning. However, the presentation is a bit hard to follow and the empirical evidence are not strong enough to support the claims. Please see below for detailed comments:

1. The writing is a bit sloppy and sometimes hard to follow. For example, in Equation (1), the right hand side "X" should probably be "X^d"? In Equation (3), tilda(W) is denoted as "the weights for network 2", but I had a hard time figuring out how is "network 2" constructed. Is it just a one layer linear classifier? What are the inputs and outputs? How is tilda(W) used?

2. Empirical results are weak: Comparison to previous ensemble learning baselines are missing. In the main text, we only see an experiments on a 2D synthetic dataset. While synthetic dataset could be quite useful to illustrate the intuitions, it could also be misleading as the behaviors in high dimensional data could be quite different. So using this synthetic example as the main experiments makes the paper less convincing.

3. This paper focus on tiny, under-parameterized networks, making it less clear how relevant the results presented here are to the literature. This paper argues that the rationale is that evaluating a large overparameterized networks could be a computational bottleneck. However, evaluating multiple under-parameterized networks with an ensembling method could also lead to extra computation overhead. It would be useful to back this argument up with an empirical comparison between single overparameterized models and ensemble of underparameterized models, for both performance and computation costs.


**Summary Of The Paper:**

This paper studies a form of ensembling of under-parameterized networks. They show that different local minima specialize distinct subset of inputs, and an algorithm called MLSUP can be used to combine them to improve the prediction. The method is empirical tested with a 1-hidden-layer network on a 2D synthetic classification dataset (and in the appendix an artificially corrupted UCI Iris flower task).

**Summary Of The Review:**

I found the presentation could be improved, and the experiment results are too weak to support the paper.

---

### Official Review · Reviewer_DYC3 · 2021-11-03

**Correctness:** 3
**Technical Novelty And Significance:** 3
**Empirical Novelty And Significance:** 2
**Recommendation:** 5
**Confidence:** 3

**Main Review:**

The paper draws inspiration from physics and studies minima of loss landscape. They show that different local minima perform well on distinct subsets of data. It’s natural to wish to combine the strengths of these local minima. How to select and combine the local minima? The authors propose a framework and selection criteria. The recommended selection criteria are interesting. The authors share insights on why these criteria are better than the alternative approach based on Euclidean distance. I think the insights are useful.

I feel it would be helpful if the authors can provide more details on how they compute the heat capacity. In section 4.3 the method description seems a bit vague. In the experiment, 7,842 minima were found,  then two were selected. The big gap may need some discussion. In the introduction, the authors mentioned low-lying and high-lying minima, and that low-lying non minima may be better than high-lying minima. There’s no further discussion in the rest of the paper, making parts of the intro seemingly disconnected.

Overall, although the proposed methods are interesting, the presentation and discussion need improvements.


**Summary Of The Paper:**

The paper proposes a method to select a subset of local minima of loss function landscape and combine the minima using a second meta-network which learns to select a set of weights for each input data point. The subset of local minima is selected based on two criteria, minima that are distant in classification space and peak heat capacity. The combination of selected minima achieves better classification accuracy than the best individual and minima selected based on Euclidean distance.


**Summary Of The Review:**

The authors adopt methods from physics and investigate local minima of  loss landscape. The methods are interesting and insights are useful. However, the current presentation is unsatisfactory and needs major improvements.  Sections of the paper seem disconnected or repetitive, and the methods sections lack details.

---

### Decision · Program_Chairs · 2022-01-20

**Decision:**

Reject

**Comment:**

All reviewers agree that the paper is below the acceptance threshold and the authors did not respond to the reviews.
In summary, this is a clear reject